# Short-Term Estivation and Hibernation Induce Changes in the Blood and Circulating Hemocytes of the Apple Snail *Pomacea canaliculata*

**DOI:** 10.3390/metabo13020289

**Published:** 2023-02-16

**Authors:** Cristian Rodriguez, Alejandra D. Campoy-Diaz, Maximiliano Giraud-Billoud

**Affiliations:** 1Instituto de Histología y Embriología de Mendoza (IHEM), Universidad Nacional de Cuyo–CONICET, Mendoza 5500, Argentina; 2Departamento de Biología, Facultad de Ciencias Exactas y Naturales, Universidad Nacional de Cuyo, Mendoza 5500, Argentina; 3Instituto de Fisiología, Facultad de Ciencias Médicas, Universidad Nacional de Cuyo, Mendoza 5500, Argentina; 4Departamento de Ciencias Básicas, Escuela de Ciencias de la Salud-Medicina, Universidad Nacional de Villa Mercedes, San Luis 5730, Argentina

**Keywords:** Oxidative stress defenses, hypometabolism, immune system, hemocyanin

## Abstract

States of natural dormancy include estivation and hibernation. Ampullariids are exemplary because they undergo estivation when deprived of water or hibernation when exposed to very low temperatures. Regardless of the condition, ampullariids show increased endogenous antioxidant defenses, anticipating the expected respiratory burst during reoxygenation after reactivation, known as “Preparation for Oxidative Stress (POS)”. In this work, we tested the POS hypothesis for changes in the blood and hemocytes of the bimodal breather *Pomacea canaliculata* (Ampullariidae) induced at experimental estivation and hibernation. We described respiratory (hemocyanin, proteins, lactate), antioxidant (GSH, uric acid, SOD, CAT, GST), and immunological (hemocyte levels, ROS production) parameters. We showed that, although the protein level remains unchanged in all experimental groups, hemocyanin increases in response to estivation. Furthermore, lactate remains unchanged in challenged snails, suggesting an aerobic metabolism during short-term challenges. Blood uric acid increases during estivation and arousal from estivation or hibernation, supporting the previously proposed antioxidant role. Regarding hemocytes, we showed that the total population increases with all challenges, and granulocytes increase during hibernation. We further showed that hibernation affects ROS production by hemocytes, possibly through mitochondrial inhibition. This study contributed to the knowledge of the adaptive strategies of ampullariids to tolerate adverse environmental conditions.

## 1. Introduction

Freshwater animals endure harsh environmental conditions, such as low oxygen or water availability or high/low temperatures, entering diverse dormant states, often of low metabolic activity, i.e., with low or zero oxygen consumption [1,2,3]. Natural dormancy states include—but do not limit to—estivation and hibernation [4]. Mollusks are notable in this respect because very many species undergo one of those two processes [5]. Besides, adverse environmental conditions can also compromise the molluscan immune system, often resulting in disease outbreaks [5].

Under natural conditions, estivating or hibernating gastropods often rely on anaerobic metabolism to obtain energy during the harsh period [6]. That could be particularly true for gastropods that rely solely on either aquatic or aerial respiration, which are the species that have received the most attention [5]. However, it may not be the case for bimodal-breathing gastropods, which rely on both gill and lung respiration. For instance, for bimodal breathing ampullariids, there is contrasting evidence regarding the utilization of oxygen during the dormant state [7,8,9]. Indeed, ampullariids are exemplars because most species can both estivate and hibernate; however, the challenge of oxygen deprivation during the stressful period might be similar [10].

Whatever the stressful condition that is going through, ampullariids may augment endogenous antioxidant defenses, anticipating the expected respiratory burst during reoxygenation upon reactivation [11,12,13]. The described pattern, namely “Preparation for Oxidative Stress (POS)”, occurs in more than 100 species distributed among nine phyla [14]. The holistic study of the mechanisms that could be involved in this pattern has recently been addressed in depth, which involves elucidating interactions at different levels [15]. The POS theory, thus, predicts that endogenous antioxidants increase when the animal attains the dormancy state [16]. In gastropods, however, the POS strategy has been mainly tested for tissue antioxidants, but little is known about changes occurring in the blood [17].

Among ampullariids, the apple snail *Pomacea canaliculata* is one of the worst invasive in the world [18]. Introduced apple snails have caused negative economic and ecological impacts [19], partly due to different biological peculiarities that allow them to settle in new habitats [10]. Among other physiological adaptations, the species estivates when the body of water it inhabits dries up and can withstand large temperature fluctuations, hibernating at times of the year when the temperature drops significantly [10]. Indeed, there are descriptions in the literature of tolerance to temperatures as low as 4 °C [20].

Previous laboratory studies have shown that estivating or hibernating apple snails adjust to the pattern of the POS strategy [11,12,13,21]. In experimentally induced estivation and hibernation periods of 45 days, tissue antioxidants such as uric acid and glutathione (GSH) increase and act as a protective mechanism [11,12]. We should consider that it is not surprising that the species utilizes uric acid as an antioxidant because a specialized tissue contains urate deposits in different organs, which gives the animal the advantage of having this molecule available when required [21,22,23]. Under short-term estivation (7 days), however, enzymatic antioxidant activities of catalase (CAT) and superoxide dismutase (SOD) augment in several tissues while that of glutathione S-transferase (GST) diminishes in the digestive gland. Moreover, the enzymatic response seems mediated by the REDOX-sensitive transcription factor forkhead box protein O 3 (FOXO3) because its expression increases in tissues exposed to an activity–estivation–arousal cycle, unlike hypoxia-inducible factor-1 alpha (HIF1α) and nuclear factor erythroid 2-related factor 2 (Nrf2), two other REDOX-sensitive transcription factors, which show decreased expression during the cycle [13]. Therefore, *P. canaliculata* apparently triggers an enzymatic defense response that may change to a non-enzymatic defense response.

Little is known about whether estivation or hibernation compromises the immune system of gastropods. Gastropod immune defenses are essentially constituted by physical barriers and innate immunity, which involves both molecular and cellular effectors [24]. The main cellular effectors of the ampullariid immune system are circulating and tissue hemocytes [25]. As such, hemocytes mediate defense responses through phagocytosis, aggregation/nodulation/encapsulation, and they may produce lytic enzymes and ROS to destroy pathogens [26,27]. In *P. canaliculata*, hemocytes can be classified into three different populations, namely hyalinocytes (~65%), granulocytes (~10%), and agranulocytes (~25%) [25]. Therefore, it is interesting to characterize changes in the blood and hemocytes of *P. canaliculata* exposed to activity–dormancy–arousal cycles induced under controlled laboratory conditions.

Herein, we evaluated respiratory (hemocyanin, proteins, lactate), antioxidant (GSH, uric acid, SOD, CAT, GST), and immune (hemocyte levels, ROS production) parameters in the blood of short-term estivated and hibernated animals. In addition, we complemented previous data by showing changes in tissue antioxidants (CAT and GST) under experimentally induced 7-day hibernation. Finally, we compared the data obtained with those from the few studies on other ampullariids.

## 2. Materials and Methods

### 2.1. Animals and Experimental Conditions

Animals from our laboratory-cultured Rosedal strain (N = 6–14 per group) were separated into groups with equal numbers of adult males and females (4–5 months old). Active (control) animals were kept in aquaria at 26–28 °C (aquarium tap water was changed three times a week) and fed ad libitum with a diet consisting of fresh lettuce, dried *P. canaliculata* eggs, and carp food pellets (Shulet Peishe, Argentina). Culture conditions have been previously described (e.g., [21]). Two experimentally induced dormant (hypometabolic) states were set: (I) animals kept out of water in plastic containers for 7 days (short-term estivation), and (II) animals kept in an aquarium at 10–12 °C for 7 days (short-term hibernation). Half of the animals from each experimental set were immersed in water at 25–28 °C for 30 min, during which they resumed activity (the “arousal” groups). Thus, animals comprised five groups: (1) control (active), (2) estivation (est), (3) arousal-est, (4) hibernation (hib), and (5) arousal-hib. Our laboratory has previously characterized the experimental inductions of activity–dormancy–arousal cycles [11,12,13,21].

### 2.2. Blood Withdrawal and Determination of Circulating Hemocyte Concentration

Blood from the heart was obtained with a syringe moistened with a buffered antiaggregant solution (*Pc*ABS), according to the method reported by Rodriguez et al. [28], and kept in microtubes on ice until use. Blood aliquots of 50 µL were properly diluted with *Pc*ABS and stained with 0.02% Trypan Blue to determine hemocyte concentrations (as the total hemocyte level) using a Neubauer hemocytometer. Additional aliquots from each animal sample were either used immediately for flow cytometry analyses (Section 2.4.1) or stored at −80 °C for further biochemical determinations, as described below.

### 2.3. Biochemical Determinations

#### 2.3.1. Total Protein Concentration

Protein content was measured by the method of Lowry et al. [29], using different bovine serum albumin concentrations as standard solutions. Protein concentration was expressed as milligrams per milliliter of blood (mg/mL).

#### 2.3.2. Hemocyanin

Hemocyanin concentration was estimated according to the method described by Behrens et al. [30]. Fresh blood was diluted 30-fold in buffer (glycine 50 mM, EDTA 10 mM, pH 8.8), and absorbance was measured with a spectrophotometer at 346 nm. The values were converted to millimoles per liter (mM) of blood, multiplying it by a correction factor of 3.28.

#### 2.3.3. Lactate

The lactic acid from the blood samples was measured using a commercial kit (Wiener Lab), following the manufacturer’s instructions. Briefly, lactate is oxidized by the enzyme lactate oxidase, and the H_2_O_2_ formed is used by peroxidase to generate a chromogen that is measured at 540 nm. Concentrations of lactic acid were expressed as micrograms per milliliter of blood (µg/mL).

#### 2.3.4. Uric Acid

Blood samples were treated with uricase, and the amount of H_2_O_2_ formed was measured at 510 nm by peroxidase-catalyzed reaction with 4-aminophenazone and chlorophenol, which produces a colored quinoneimine product [31]. The uric acid concentration was expressed as a milligram of urate per milliliter of blood (mg/mL).

#### 2.3.5. Reduced Glutathione (GSH)

GSH concentration was determined by the method described by Beutler et al. [32]. Reduced glutathione hydrogen sulfide compounds develop a stable color when 5,5-dithio-bis-2-nitrobenzoic acid (DTNB) is added to them, and it can be measured at a wavelength of 412 nm. The concentration was expressed as a microgram of reduced glutathione per milliliter of blood (µg/mL).

#### 2.3.6. Enzymatic Activities

Antioxidant enzyme activities were spectrophotometrically quantified in blood samples (SOD = 50 µL, CAT = 20 µL, GST = 50 µL) and in animal tissues from the experimental groups. Around 100 mg of frozen samples from the digestive gland, the gill, and the lung of control, hibernation, and arousal-hib animals were homogenized (UltraTurrax^®,^, IKA Werke, Staufen, Germany) in buffered saline solution and centrifuged (10,500× *g* at 4 °C for 5 min). Supernatants were collected, aliquoted, and frozen for the determination of enzymes and proteins.

SOD activity was determined according to Misra and Fridovich [33]. The inhibition of the auto-oxidation of epinephrine was measured at 480 nm (30 °C), and a unit (U) was defined as the amount of SOD that inhibits 50 % of adrenochrome formation. The activity of SOD was expressed as Units per milligram of protein (U/mg). CAT activity was measured by the method described by Aebi [34] through the decomposition of 10 mM H_2_O_2_ by the sample in 50 mM phosphate buffer (pH 7.0). The decrease in the absorbance of H_2_O_2_ for 60 s at 240 nm represents the enzyme activity, and it was expressed as Units of CAT per milligram of protein (U/mg). GST activity was calculated by the Habig et al. [35] method, where the increase in the absorbance for 180 s at 340 nm from a mixture of buffer solution (20 mM Tris base, 1 mM EDTA, 1 mM dithiothreitol, 0.5 M sucrose, 0.15 M KCl, and 0.1 mM phenylmethylsulfonyl fluoride; pH 7.6), 50 mM 1-chloro-2,4-dinitrobenzene, 100 mM reduced glutathione, and the sample. Results of GST activity were expressed as milliUnits per milligram of protein (mU/mg).

### 2.4. Intracellular ROS Production

#### 2.4.1. Hemocyte Exposure to DCF

Blood samples (200–1000 µL) from each animal were diluted (1:2) with *Pc*ABS and centrifuged (700× *g*) at 4 °C for 10 min, and the hemocyte pellets were suspended in 100 µL of *Pc*ABS. Samples were incubated with 2′,7′-dichlorofluorescin-diacetate (DCFH-DA; D6883, Sigma-Aldrich) at a final concentration of 10 μM to assess ROS production by the hemocytes from each studied group. DCFH-DA is a membrane permeable non-fluorescent molecule that enters cells readily and is oxidized to the impermeable and fluorescent product 2′,7′-dichlorofluorescin (DCF), mainly by H_2_O_2_-Fe^2+^-derived oxidant but also, to a lesser extent, by other reactive intermediates [36]. Thus, the DCF mean fluorescence can be used as an indicator of overall ROS production [37,38]. All samples were incubated in the dark at room temperature (25 °C), centrifuged (700× *g*) at 4 °C for 10 min, and washed with 1 mL *Pc*ABS before analysis by flow cytometry.

#### 2.4.2. Flow Cytometry: Hemocyte Gating and Analyses

Total hemocytes were analyzed under a FACSAria III flow cytometer (Becton–Dickinson Bioscience, San Jose, CA, USA) according to size and complexity by using pseudocolor plots of forward light scatter (FSC) and side light scatter (SSC). Hemocyte subpopulations were classified as described by Cueto et al. [27]: agranulocytes (agr), granulocytes (gra), and hyalinocytes (hya). DCF fluorescence (arbitrary units) was recorded by the FL1 detector of the flow cytometer (530/30 nm bandpass). Thus, results were expressed as cell cytograms indicating the size (FSC value), complexity (SSC value), and DCF fluorescence level (FL1 value).

DCF histograms were used to visualize and identify distinct peaks of ROS production in total hemocytes. The geometric mean of DCF fluorescence (GeoMean ×10^3^) was used to quantify intracellular ROS production. Because half of the control hemocytes (%DCF^+^ = 47.2 ± 18.1) showed elevated levels of spontaneous ROS production (Geo Mean = 2.1 ± 0.43), we classified the two DCF subpopulations (Appendix A) as DCF^−^ (MFI < 10^3^) and DCF^+^ (MFI > 10^3^). For further analyses, only the DCF^+^ subpopulations were considered for each animal and used to determine differences in MFI between groups. Samples without DCF were used to set hemocyte autofluorescence levels, which were then excluded from further analyses. A minimum of 10,000 events were recorded for each sample. All flow cytometry analyses were performed using FlowJo^®^ v.7.6 software (Becton, Dickinson and Company, Franklin Lakes, NJ, United States).

### 2.5. ROS Inhibition by CCCP

Data from bivalves have shown that various sources account for intracellularly generated ROS in circulating hemocytes [37]. Therefore, we assessed this possibility by inhibiting mitochondria-generated ROS with the membrane potential disruptor carbonyl cyanide-3-chlorophenylhydrazone (CCCP), which has already been used for *P. canaliculata* hemocytes [28]. Thus, a replicate of each sample treated with DCF was simultaneously exposed to CCCP at a final concentration of 10 μM, as an inhibitor of mitochondrial ROS production [39]. The inhibition was calculated as the proportion between the geometric mean of total DCF fluorescence of challenged hemocytes and those inhibited by CCCP [38]. Results were expressed for both total hemocytes and separate hemocyte subpopulations.

### 2.6. Statistics

Data were checked for normality with the Shapiro–Wilk test. As some data did not fit a normal distribution, further analyses were made using non-parametric tests. Kruskal–Wallis test and Dunn’s multiple comparisons were carried out to compare each variable of est, arousal-est, hib, and arousal-hib animals vs. control active animals (GraphPad Prism^®^ v.8.0 for Windows, GraphPad Software, San Diego, CA, USA). Differences were considered significant at *p* < 0.05. The results are expressed as means ± standard errors (SEM).

## 3. Results

### 3.1. Changes in the Blood Induced by Activity–Dormancy–Arousal Cycles 

The blood of *P. canaliculata* showed some modification during the experimental conditions imposed during activity–dormancy–arousal cycles (Figure 1 and Table 1).

Total protein concentration did not show significant changes in the blood of estivating or hibernating animals and the corresponding arousal states (Figure 1a and Table 1). On the other hand, hemocyanin showed a significant increase during estivation compared with active animals (0.65 ± 0.07 mM/L vs. 0.40 ± 0.03 mM/L, respectively), and hemocyanin concentration remained high after the arousal (0.53 ± 0.04 mM/L; Figure 1b and Table 1). Nevertheless, hibernation did not induce significant changes in the exposed animals (Figure 1b and Table 1). Lactate levels did not show any significant change during the activity–dormancy–arousal cycles, but a decrease was evident only in the hib group (Figure 1c and Table 1). Contrarily, uric acid levels significantly increased during est compared with active animals (12.1 ± 3.6 vs. 5.3 ± 0.3 mg/mL, respectively), and this figure remained elevated in the arousal-est group (7.2 ± 0.5 mg/mL; Figure 1d and Table 1). During hibernation, the urate concentration showed a tendency to increase, but this change was not statistically significant (Figure 1d and Table 1). Reduced glutathione concentrations did not change significantly during the activity–estivation–arousal cycle; however, the levels were increased in the blood of hib and arousal-hib animals, being significantly higher than the active group in aroused animals (Figure 1e and Table 1).

SOD activity showed a tendency to increase in the estivation and arousal-est groups, but no significant changes were observed during the activity–dormancy–arousal cycles (Figure 1f and Table 1). CAT showed a significant drop in its activity in the blood of hibernating animals compared with controls (0.5 ± 0.2 vs. 1.5 ± 0.1 U/mg, respectively), although, in estivation and arousal of both dormant states, the activity levels were lower than in the control group (est = 1.2 ± 0.2, arousal-est = 1.1 ± 0.2, arousal-hib = 0.9 ± 0.2 U/mg; Figure 1g). Under our experimental conditions, GST only showed a non-significant increase in its activity during the arousal of estivation and hibernation (Figure 1h and Table 1).

### 3.2. Changes in Circulating Hemocytes

The level and composition of circulating hemocytes showed modifications in challenged animals as compared with the control group (Figure 2). The hemocyte level increased by six-fold for the est group (*p* < 0.05), and this level remained high in the arousal-est group (Figure 2a). Contrastingly, the hemocyte level increased only by two-fold for hib animals, which was not significant (*p* = 0.74) compared with the control group, but that increased significantly by four-fold for the arousal-hib group (*p* < 0.05). Different hemocyte subpopulations showed changes in their relative frequencies only in the hib and arousal-hib groups (Figure 2b). The percent of granulocytes was higher only in hib animals, while that of hyalinocytes was higher in both hib and arousal-hib animals, compared with control ones. The percent of agranulocytes showed no changes between groups.

### 3.3. Production of ROS by Circulating Hemocytes

Figure 3a shows representative DCF histograms—indicative of ROS levels—of hemocytes from the control and challenge groups, and Figure 3b shows the quantification of the observed changes. Est and arousal-est hemocytes showed a distinct peak at 10^4^ (Geo Mean = 2.4 ± 0.7 and 2.0 ± 0.83, respectively), but this did not differ from active hemocytes (Geo Mean = 2.1 ± 0.43). For hib hemocytes, the overall DCF population showed a left shift of the 10^4^-peak, which was quantitatively evidenced as a decrease in the DCF^+^ MFI (Geo Mean = 1.2 ± 0.26). ROS levels remained low in arousal-hib hemocytes (Geo Mean = 1.0 ± 0.31), which also showed an overall left-shift of the 10^4^-peak.

### 3.4. ROS Inhibition by CCCP

The inhibition of ROS production by CCCP was analyzed for both total hemocytes and the different subpopulations, i.e., agranulocytes, granulocytes, and hyalinocytes (Figure 4 and Appendix A). Hemocytes from the est and arousal-est groups showed little response to CCCP, i.e., no significant differences were found compared with that of the control (Figure 4a). However, the production of ROS of hib hemocytes was significantly inhibited by CCCP (*p* < 0.05), and this response remained in hemocytes from the arousal-hib group. When analyzing the subpopulations separately, all of these showed the same response for the hib and arousal-hib groups (Figure 4b).

## 4. Discussion

The dormant state induced by estivation or hibernation represents a challenging hypometabolic condition that can affect the oxidative balance in those animals that must resume normal metabolic activity in the arousal [1,2,3]. The apple snail *P. canaliculata* shows notable adaptations, such as the possibility of estivating and hibernating to tolerate seasonal activity–dormancy–arousal cycles [10]. In this way, the POS strategy is used by this species to avoid damage to vital cellular components by free radicals in tissues [13,15]. Until now, in those gastropods where the use of the POS strategy has been described (i.e., reported ten species, [11,12,13,40,41,42,43,44,45,46,47,48,49,50,51,52,53,54]), the experimental tests have been carried out on tissue extracts, not taking into account the changes occurring in compounds and cells from the blood. In this study, we described, for the first time in a freshwater gastropod under laboratory conditions of estivation and hibernation, modifications in blood compounds and antioxidants, and changes in the circulating hemocytes.

During dormant states, proteins may be catabolized, resulting in a decrease in the circulating proteins, as has been described for hibernating *Helix pomatia* snails [55]. Nevertheless, in the two activity–dormancy–arousal cycles explored in our experiments, the total protein concentration remained unchanged, suggesting that protein turnover was at a steady state, as observed for the abalone *Haliotis midae* exposed to hypoxia [56].

In ampullariids, the utilization of oxygen during estivation shows variations. For instance, Prashad [7], Coles [8], and Burky et al. [9] have respectively shown that *Pila globosa*, *Pila ovata*, and *Pomacea urceus* utilize oxygen stored in the lung during estivation. However, Meenakshi [57,58] has found that *Pila virens* respires anaerobically during prolonged estivation (>6 months). Herein, we showed that circulating lactate levels, as an indicator of anaerobic metabolism, do not change in challenged animals, which suggests that, during the first week of estivation or hibernation, the obtention of energy would still be aerobic for *P. canaliculata*. For estivating animals, this is paralleled by the increase in hemocyanin levels, which based on its biochemical properties at this condition (i.e., a positive Bohr effect at pH values between 6.5 and 9; [59]), would still deliver oxygen, thus maintaining aerobiosis. Therefore, the observed increase could be due to the upregulation of hemocyanin genes [60]. Likewise, in crustaceans exposed to hypoxic conditions, hemocyanin increases its concentration, a response mediated by HIF-1α [61,62].

There is no previous data on the utilization of oxygen during hibernation in ampullariids. Herein, the hibernation group showed no change in either hemocyanin or lactate levels (Figure 1). However, as our experimental hibernation cycle takes place underwater and only lasts seven days, we postulate that oxygen exchange continues through the lung, acting as a physical gill, as occurs when some lung-bearing hygrophilan taxa do not have access to air [5]. Together, our data agree with an aerobic metabolism in blood during both short-term estivation and hibernation periods.

The short-term activity–dormancy–arousal cycles induced in our study showed in *P. canaliculata* an increase in non-enzymatic antioxidant defenses (uric acid) in the blood, as it is observed in tissues after long periods of estivation or hibernation [11,12]. However, GSH does not seem to play an important role in the blood during both hypometabolic states, unlike what occurs in tissues after long periods of estivation or hibernation, because we observed a significant change in GSH concentration only in the arousal from hibernation. Likewise, enzymatic antioxidants do not seem to have an important role in the blood in the face of possible imbalances of oxyradicals since CAT activity was lower than those of the blood of control animals during hibernation, while SOD and GST activities remained without significant changes (see Figure 1).

The POS strategy involves the participation of antioxidants that rise during metabolic depression [15,63]. As previously described, during short periods of estivation (7 days), an increase in SOD and CAT activities was observed in several tissues [13]. Meanwhile, after 7 days of hibernation, animals showed a significant increase in the activity of CAT in the digestive gland and of GST in the gill and lung (Table 2). GST activity is still significantly higher in all studied tissues of aroused animals compared to the control group, an issue that could be related to the detoxification role that this enzyme may have to fulfill to eliminate compounds generated after the reactivation of cellular function (Table 2).

Only a few reports have explored the effect of estivation and hibernation on the immune system of freshwater gastropods. The report of Bhunia et al. [64] for *Pila globosa* shows that total hemocyte levels increase after 15 days of estivation and arousal. On the other hand, hibernating *Helix pomatia* exhibits a reduction in the percentage of hyalinocytes, granulocytes, and agranulocytes [65]. In this study, circulating hemocyte levels tended to increase in all challenged animals (see Figure 2a), although significance was not reached for hibernating animals. However, the latter exhibited changes in the composition of the circulating hemocyte population (see Figure 2b). In hibernating animals, granulocytes increased, and hyalinocytes concomitantly decreased compared to control animals. Animals arousing from hibernation followed a similar trend, but this was only significant for the drop in hyalinocytes.

The observed increase in hemocyte levels is higher than that expected due to water loss during short-term estivation (10-fold vs. 5-fold; [21]); therefore, our results support the hypothesis that Bhunia et al. [64] suggested for *Pila globosa* of “hyper mobilization” of hemocytes from tissue reservoirs, as it may be the case for the kidney of *P. canaliculata* [26]. Notwithstanding, we may also consider that the circulating hemocytes of *P. canaliculata* can rapidly divide in the blood within six days [28], which also may explain the high hemocyte levels found in challenged animals.

Almost half of the circulating hemocytes from control animals showed high spontaneous ROS production (Figure 3a). This is analogous to what Lambert et al. [66] already reported for hemocytes of the Pacific oyster *Crassostrea gigas* in the absence of any stimulation. Indeed, in bivalves, different sources account for ROS production in hemocytes [67].

ROS production by circulating hemocytes was different between estivating and hibernating animals. In estivating animals, the ROS production of hemocytes tended to increase compared to controls, but this was not significant, which agrees with our result from the biochemical composition of the blood (see Figure 1), which together suggests that aerobic metabolism is still ongoing in the short-term estivation challenge. However, ROS production decreased nearly two-fold in hemocytes from hibernating animals and those arousing from hibernation. As the flow cytometry histograms (Figure 3a and Appendix A) show, hibernation-induced circulating hemocyte changes appear towards a unimodal population. This suggests that besides the low levels of ROS production found, the two subpopulations classified as DCF^−^ and DCF^+^ responded differently to the hibernation challenge.

Further analysis showed that total ROS production, measured as the decrease in the geometric mean of DCF fluorescence, could be inhibited significantly by the mitochondrial membrane potential disruptor CCCP (as an inhibitor of the mitochondrial ROS source) in hemocytes from hibernating animals and those reactivating form hibernation (Figure 4a).

Using a similar approach, Donaghy et al. [37] have reported that, in *C. gigas* hemocytes, the production of ROS originates mainly from mitochondria. However, recently, Kajino et al. [68] showed that, in addition to mitochondria, lysosomes are prime sites of ROS production in marine gastropods. This agrees with the concomitant increase in granulocytes and decreases in hyalinocytes in hibernating animals, which together increase the number of lysosomes in the total hemocyte population, since granulocytes possess large numbers of lysosome-like granules in their cytoplasm (25, 27). Thus, the inhibition of mitochondrial ROS by CCCP results in only the production of lysosomal ROS, which is observed as a “shift to the left” in the DCF histograms. We then analyzed whether the differences already found for the hemocyte subpopulations accounted for the differences observed regarding total ROS production, but all the hemocyte subpopulations showed a similar response (Figure 4b). Therefore, future studies using imaging techniques to visualize ROS probes in mitochondria or lysosomes would be required to reveal the main site of ROS production in *P. canaliculata* hemocytes induced at different activity–dormancy–arousal cycles.

After an overview of the results obtained in this work and previous reports related to the physiological adaptations of ampullariids, it is interesting to consider the possible interaction of different regulatory factors, which can be assessed both in vitro and in vivo in this experimental model and could open new fields of knowledge that allow us to understand better the different adaptive strategies of animals to tolerate adverse environmental conditions.

## Figures and Tables

**Figure 1 metabolites-13-00289-f001:**
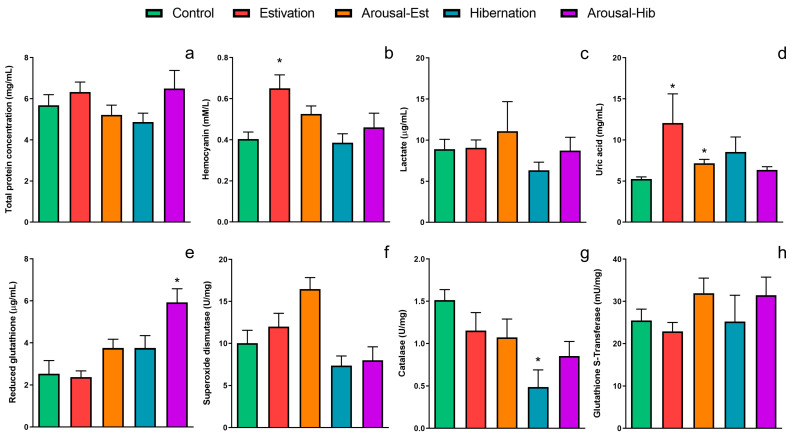
Biochemical compounds and antioxidant enzymes in the blood of *P. canaliculata* during activity–dormancy–arousal cycles. (**a**) Total protein concentration; (**b**) hemocyanin; (**c**) lactate; (**d**) uric acid; (**e**) reduced glutathione; (**f**) superoxide dismutase; (**g**) catalase; and (**h**) glutathione S-transferase activity. Mean ± SEM. Asterisks (*) indicate significant differences with the control group (*p* < 0.05, Kruskal-Wallis, Dunn’s test).

**Figure 2 metabolites-13-00289-f002:**
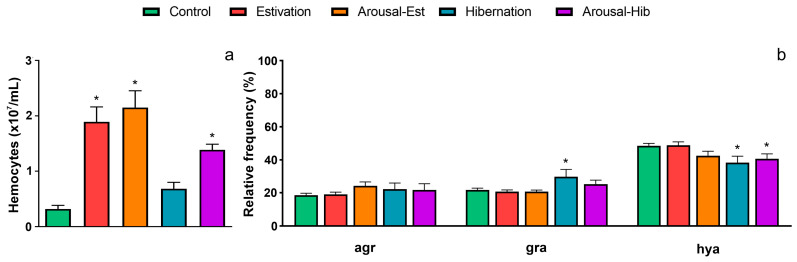
Changes in the circulating hemocytes of challenged animals. (**a**) Total hemocyte levels; (**b**) relative composition of circulating hemocytes. Asterisks (*) indicate significant differences with the control group (*p* < 0.05, Kruskal-Wallis, Dunn’s test).

**Figure 3 metabolites-13-00289-f003:**
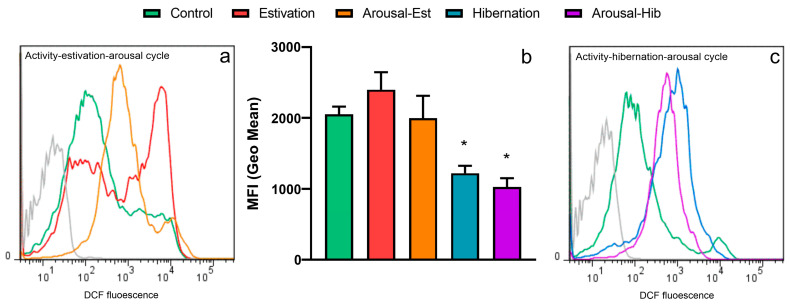
Production of ROS by circulating hemocytes. (**a**) Flow cytometry histograms showing distinct peaks of DCF fluorescence corresponding to different ROS levels for active, est, and arousal-est hemocytes; (**b**) changes in the mean fluorescence intensity (MFI) of DCF between groups; (**c**) histograms showing the fluorescence peaks of active, hib and arousal-hib hemocytes. Gray histograms represent autofluorescence controls. Asterisks (*) indicate significant differences with the control group (*p* < 0.05, Kruskal-Wallis, Dunn’s test).

**Figure 4 metabolites-13-00289-f004:**
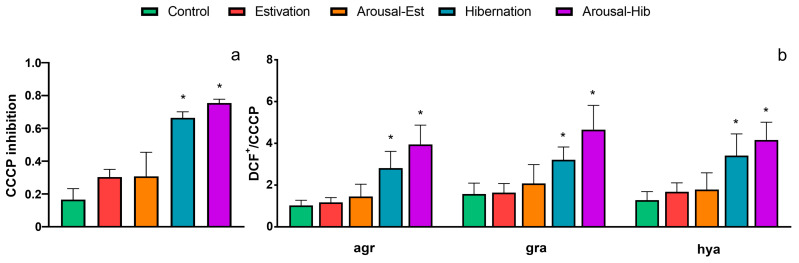
Inhibition of ROS production by CCCP. (**a**) Inhibition ratio for total hemocytes; (**b**) inhibition ratio for separate hemocyte subpopulations. Abbreviations: agr, agranulocytes; gra, granulocytes; hya, hyalinocytes. Asterisks (*) indicate significant differences with the control group (*p* < 0.05, Kruskal-Wallis, Dunn’s test).

**Table 1 metabolites-13-00289-t001:** Concentrations of proteins (mg/mL), hemocyanin (mM/L), lactate (µg/mL), uric acid (mg/mL), reduced glutathione (µg/mL) and enzyme activity of superoxide dismutase (SOD) (U/mg), catalase (CAT) (U/mg) and glutathione S-transferase (GST) (mU/mg) in hemolymph of animals exposed to activity–dormancy–arousal cycles.

	Control	Estivation	Arousal-Est	Hibernation	Arousal-Hib
Proteins	5.7 ± 0.5	6.3 ± 0.5	5.2 ± 0.5	4.9 ± 0.4	6.5 ± 0.9
Hemocyanin	0.40 ± 0.03	0.65 ± 0.06 *	0.53 ± 0.04	0.39 ± 0.04	0.46 ± 0.07
Lactate	8.9 ± 1.2	9.1 ± 0.9	11.1 ± 3.6	6.3 ± 1.0	8.7 ± 1.6
Uric Acid	5.3 ± 0.3	12.1 ± 3.6 *	7.2 ± 0.5 *	8.5 ± 1.8	6.4 ± 0.4
GSH	2.5 ± 0.6	2.4 ± 0.3	3.8 ± 0.4	3.8 ± 0.6	5.9 ± 0.6 *
SOD	10.0 ± 1.5	12.0 ± 1.6	16.5 ± 1.4	7.4 ± 1.1	8.0 ± 1.6
CAT	1.5 ± 0.1	1.2 ± 0.2	1.1 ± 0.2	0.5 ± 0.2 *	0.9 ± 0.2
GST	25.5 ± 2.7	22.9 ± 2.1	31.9 ± 3.6	25.2 ± 6.2	31.4 ± 4.3

* Indicates significant differences vs. control group (Kruskal-Wallis, Dunn’s tests; *p* < 0.05). Mean ± SEM.

**Table 2 metabolites-13-00289-t002:** Antioxidant enzymes activity in tissues of *Pomacea canaliculata* exposed to short-term activity-hibernation-arousal cycle. CAT: catalase; GST: glutathione S-transferase. Mean ± SEM.

	Control	Hibernation	Arousal
**CAT**			
Gill	9.8 ± 1.5	14.0 ± 1.7	13.8 ± 4.1
Digestive gland	65.8 ± 18.9	341.3 ± 94.8 *	187.0 ± 39.7
Lung	15.2 ± 2.9	7.8 ± 3.4	5.5 ± 0.5
**GST**			
Gill	89.8 ± 29.5	471.5 ± 52.2 *	1118.0 ± 126.6 *
Digestive gland	168.0 ± 53.8	848.7 ± 197.7	1538.0 ± 291.7 *
Lung	65.2 ± 7.4	338.4 ± 151.9 *	961.1 ± 179.5 *

Asterisks (*) indicate significant differences with the control group (*p* < 0.05, Kruskal-Wallis, Dunn’s test).

## Data Availability

Not applicable.

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
