# Peer review of "Short-Term Estivation and Hibernation Induce Changes in the Blood and Circulating Hemocytes of the Apple Snail Pomacea canaliculata"

_metabolites, 2023, doi:10.3390/metabo13020289_

Round 1

Reviewer 1 Report

Manuscript ID:  metabolites-2178057       Submission Type:  Article

Title:  Short-term estivation and hibernation induce changes in the blood and circulating hemocytes of the apple snail Pomacea canaliculate

The manuscript entitled of “Short-term estivation and hibernation induce changes in the blood and circulating hemocytes of the apple snail Pomacea canaliculate” submitted by Cristian Rodriguez et al., to Metabolites, aiming to firstly describe changes of concentrations of 3 compounds (hemocyanin, lactate, urate) and activities of 2 antioxidant enzymes (CAT & GST) in the blood of apple snail Pomacea canaliculate during “activity-estivation- hibernation-arousal” cycles, and secondly to assess the hemocyte response to challenges contemplating modifications in subpopulations.

Overall, this manuscript was not very well prepared, and the data needed to be nicely presented. The supporting materials needed to move back to the main text in order to enhance the data. Otherwise, the presented data were rather weak to support the title and the conclusion. The significance of this study should be pointed out in a much clear way. Besides, there are quite some concerns remained to be fully addressed. I would suggest a careful improvement can be made on this manuscript. Substantial revision is herein required for this submitted manuscript.

Major issues:

1.      Only description of phenomena and observations, in this story, meaning the changes of compounds’ concentrations and enzymes’ activities, and of one more assessment of hemocyte response to stress and challenges, but lacking of mechanism for changes and responses, or failing to summary the major impressive finding and advancements in science, would be not enough to be published on any high-quality international journals.

2.      The author reported the changes of antioxidant enzymes (CAT, GST) upon exposures to short-term (7 d) activity-estivation/hibernation–arousal cycles. However, the authors did not mention why those two enzymes were chosen and how many other antioxidant enzymes in the apple snail. The importance and proposed mechanism of CAT and GST for snail physiological activity or pathological function should be mentioned in the manuscript. Any introduction for pointing out the role of snail antioxidant systems and some explanation for not choosing other enzyme system should be given.

3.      The Abstract should be rewritten. Regarding the research background, the necessities for performing this study was not mentioned. The applicable value and scientific meaning of conducting this study using this specie of apple snail were not clear. The readers will not be easily satisfied with the list of concentration measurement and observation description. People always want to know more under the phenomena.

4.      The “Keywords” section, should be properly revised. When I check up the keywords, these select words should reflect the most important scientific terms used in the study. Avoid repetition.  Avoid to use the useless keyword. When the authors picked up the key words, they did not care what they had talked about in the manuscript. Some true keywords are missing.

5.      The first paragraph in the “Introduction”, the description was just too spectacular to be true. Any direct link to the title of the manuscript or the major results or conclusions? I also checked up the fist paragraph in the “Discussion” section and found the same type of problems. The limited inadequate data cannot support such a huge goal or global view. This article should be well focused on the detailed and specific research subject in deed.

6.      Line 28, “report” should be revised to “reports”.

7.      Line 42, the first sentence should be modified. Why not start with “Apple snail Pomacea canaliculate …”? Not mentioning the apple snail first but using the Latin name instead, seems not friendly to readers.

8.      The quantifications of the antioxidant enzymes were not included in this study. I would suggest the authors to determine the transcription of antioxidant enzyme genes using qPCR. The protein expression levels of the antioxidant enzymes were not well studied using native PAGE or SDS PAGE. If the appropriate antibody against the snail CAT and GST can be commercially available, however, the western blot analysis should be added to give relative full-frame picture for this study and in fact improve the quality of data in this paper.

9.      “2.4. Intracellular ROS production and inhibition by CCCP” was rather tedious. It should be separated into two subtitles:”2.4. Intracellular ROS production” and “2.5. Inhibition by CCCP” or other suitable subtitles. This part must be modified and revised. From Line 142 to Line 182, There were “2.4. Intracellular ROS production and inhibition by CCCP” together with “2.4.1. Hemocyte exposure to DCF and flow cytometry” and “2.4.1. Hemocyte gating and flow cytometry analyses” underneath. I feel chaos when I read the texts. Quite confusing.

10.   In the Figure 1 and legends to figures, “Protein concentration” refers to “total Protein concentration”. This must be clear.

11.   Figure 1 was too easy to see, and no need to put the whole Figure 1 so large. Figure 1 occupied too much page space! Figure 1 should be re-formatted. Since the information content was so low, why the authors plotted this figure with 6 panels like such too large image? I may suggest the authors to put three panels in the up-row and three in the down-row. It is truly No Use to put the Figure 1 so big.

12.   Figure 2a was too big, but the Fig. 2b was small for each column and short in its X-axis. Could the authors make 2a and 2b of the similar column-size, meanwhile make the X-axis of 2b longer than 2a?

13.   Figure 3a was so small while Fig. 3b was of low-information but too big.

14.   Figure 4a was too big, but the Fig. 4b was small for each column and short in its X-axis. Could the authors make 4b and 4a of the similar column-size, and make the X-axis of 4b longer than 4a?

15.   Indeed, Vc and Ve and NAC, are all well-known antioxidants. Have the authors tried to use these antioxidants in the snail experiments?   

16.   In my view, I guess that SOD, glutathione level, and metabolite compounds upon ROS or OS, may also be interesting to be measured in the apple snail to draw a better conclusion, in the same experimental conditions.

17.   In terms of the ROS production as the authors determined, what kinds of ROS were detected in the apple snail? More details?

18.   Why HIF-1alpha and NRF2 were discussed but excluded in the experiments?  

19.   Line 241-245. Such as “Geo Mean = 1.0 ± 0.31 ×103”, regarding the numbers and digitals, I am confused.

20.   “…preparation for oxidative stress (POS)”. OS stands for oxidative stress, maybe O.K. Well, I do not think it is necessary to have “POS” abbreviated for this phrase.  

21.   All “table S1” should be “Table S1”. And the only Table S1 should move from the SI to the main text.

22.   There was no graphical abstract (GA) to give a visible summary for the whole study. GA is hereby requested.

I won’t agree to accept the current manuscript unless the substantial revision can be made and a clear implication of the study can be provided. My recommend for this manuscript is: Major Revision.

Author Response

Dear Editor,
We thank the reviewers for their comments on the manuscript. We have edited the manuscript to address their concerns as much as possible. The revised version underwent major reorganization, especially the Introduction and Discussion. Therefore, the reviewers’ comments may refer to paragraphs that no longer exist. In our point-by-point responses, we refer to line numbers of the new version.

In addition to the recommended changes, we have made some minor modifications (spelling, grammar, etc.) that are not indicated in the new version.

We have included in the submission a manuscript with the tracking changes tool, so that reviewers can quickly view the modifications.

We hope that our manuscript is now suitable for publication in Metabolites.

Best regards,

Dr. Maximiliano Giraud-Billoud

CONICET-UNCuyo-UNViMe

Reviewer 1

Major issues:

- Only description of phenomena and observations, in this story, meaning the changes of compounds’ concentrations and enzymes’ activities, and of one more assessment of hemocyte response to stress and challenges, but lacking of mechanism for changes and responses, or failing to summary the major impressive finding and advancements in science, would be not enough to be published on any high-quality international journals.
This is a descriptive study that aims to complement previous studies on oxidative stress in two dormancy states (estivation and hibernation) of the apple snail Pomacea canaliculata. Please note that, in gastropods, oxidative stress has been mainly tested for tissue antioxidants, but only a few studies have addressed the changes occurring in the blood. Furthermore, we show, for the first time for a caenogastropod, changes in the ROS response of hemocyte in the context of experimental hibernation. We are well aware that this is a primary data report, but we believe it may provide the basis for hypothesis to pave the way for further experimental physiological studies addressing mechanisms like the ones we have recently published [1]. Furthermore, the data provided by this work also support evidence to deepen the knowledge of the mechanisms involved in the strategy of “preparation for oxidative stress” in animals under hypometabolic states, since the proposed experimental model allows the evaluation of different cellular and molecular responses under different types of stress such as estivation and hibernation, models developed by our laboratory, as well as responses can be explored with other stressors in vivo, such as changes in salinity [2], in addition to many others in vitro models with hemocytes or other primary tissue cultures of Pomacea canaliculata. It is for all of the above that we believe that this work should be published in a high-quality journal such as Metabolites.

- The author reported the changes of antioxidant enzymes (CAT, GST) upon exposures to short-term (7 d) activity-estivation/hibernation–arousal cycles. However, the authors did not mention why those two enzymes were chosen and how many other antioxidant enzymes in the apple snail. The importance and proposed mechanism of CAT and GST for snail physiological activity or pathological function should be mentioned in the manuscript. Any introduction for pointing out the role of snail antioxidant systems and some explanation for not choosing other enzyme system should be given.

This work aims to evaluate respiratory (hemocyanin, proteins, lactate), antioxidant (GSH, uric acid, SOD, CAT, GST), and immune (hemocyte levels, ROS production) parameters in the blood of short-term estivated and hibernated animals. In addition, we complement previous data by showing changes in tissue antioxidants (CAT and GST) under experimentally induced 7-day hibernation. We have made this clear in the Introduction (lines 95-99).

The knowledge that we have today regarding this experimental model allows us to know that the cellular pathways of enzymatic antioxidant defence are like those of other animals (outlined in two previous publications [3,4]), with the exception that this particular species has uric acid deposits, which it uses as an non-enzymatic antioxidant that complements the protective mechanisms against oxidative stress. Anyway, in the revised version of the manuscript, reduced glutathione (GSH) and superoxide dismutase (SOD) were determined to complement the results from both the enzymatic and non-enzymatic systems.

- The Abstract should be rewritten. Regarding the research background, the necessities for performing this study was not mentioned. The applicable value and scientific meaning of conducting this study using this specie of apple snail were not clear. The readers will not be easily satisfied with the list of concentration measurement and observation description. People always want to know more under the phenomena.
The Abstract was rewritten following the reviewer’s recommendations.

- The “Keywords” section, should be properly revised. When I check up the keywords, these select words should reflect the most important scientific terms used in the study. Avoid repetition.  Avoid to use the useless keyword. When the authors picked up the key words, they did not care what they had talked about in the manuscript. Some true keywords are missing.
The “Keywords” section was properly revised. Thank you for noticing that.

- The first paragraph in the “Introduction”, the description was just too spectacular to be true. Any direct link to the title of the manuscript or the major results or conclusions? I also checked up the fist paragraph in the “Discussion” section and found the same type of problems. The limited inadequate data cannot support such a huge goal or global view. This article should be well focused on the detailed and specific research subject in deed.
The Introduction and Discussion have been reorganized. Some parts were rewritten for clarity and conciseness and appear highlighted in yellow in the new version. Overall, we tried to adjust the text to make clear the focus of our specific research subject. We hope it will satisfy the reviewers and the readers.

- Line 28, “report” should be revised to “reports”.
The Abstract was rewritten and this recommendation is not necessary.

- Line 42, the first sentence should be modified. Why not start with “Apple snail Pomacea canaliculate…”? Not mentioning the apple snail first but using the Latin name instead, seems not friendly to readers.
The Introduction has been rewritten. However, we have followed the recommendation in other corresponding parts (lines 3, 61, 62, 68, 328).

- The quantifications of the antioxidant enzymes were not included in this study. I would suggest the authors to determine the transcription of antioxidant enzyme genes using qPCR. The protein expression levels of the antioxidant enzymes were not well studied using native PAGE or SDS PAGE. If the appropriate antibody against the snail CAT and GST can be commercially available, however, the western blot analysis should be added to give relative full-frame picture for this study and in fact improve the quality of data in this paper.

The biochemical techniques that we used aim to determine enzymatic activity (not concentration). Enzyme expression levels or concentration may not reflect the actual status of the antioxidant system because, even though a particular enzyme is expressed, many factors can alter its enzymatic activity (as shown for hibernating turtles, Tang et al. [5]; or shrimps, Estrada-Cárdenas et al. [6]). Moreover, monoclonal antibodies against P. canaliculata SOD, CAT and GST are not currently available. Bioinformatic analysis of ampullariid gene database to construct antibodies (or qPCR specific primers) would be needed, and all that goes well beyond the aim of this paper. Probably in the future we can explore these possibilities for new publications.

- “2.4. Intracellular ROS production and inhibition by CCCP” was rather tedious. It should be separated into two subtitles:”2.4. Intracellular ROS production” and “2.5. Inhibition by CCCP” or other suitable subtitles. This part must be modified and revised. From Line 142to Line 182, There were “2.4. Intracellular ROS production and inhibition by CCCP” together with “2.4.1. Hemocyte exposure to DCF and flow cytometry” and “2.4.1. Hemocyte gating and flow cytometry analyses” underneath. I feel chaos when I read the texts. Quite confusing.
The reviewer was right. We apologize for the confusing texts. The entire section has now been reorganized and rewritten for clarity (lines 181-225).

- In the Figure 1 and legends to figures, “Protein concentration” refers to “total Protein concentration”. This must be clear.
Changed as suggested (line 264).

- Figure 1 was too easy to see, and no need to put the whole Figure 1 so large. Figure 1 occupied too much page space! Figure 1 should be re-formatted. Since the information content was so low, why the authors plotted this figure with 6 panels like such too large image? I may suggest the authors to put three panels in the up-row and three in the down-row. It is truly No Use to put the Figure 1 so big.
The recommendation was followed (line 262).

- Figure 2a was too big, but the Fig. 2b was small for each column and short in its X-axis. Could the authors make 2a and 2b of the similar column-size, meanwhile make the X-axis of 2b longer than 2a?
The recommendation was followed (line 289).

- Figure 3a was so small while Fig. 3b was of low-information but too big.
Figure 3 was properly changed (line 302).

- Figure 4a was too big, but the Fig. 4b was small for each column and short in its X-axis. Could the authors make 4b and 4a of the similar column-size, and make the X-axis of 4b longer than 4a?
The recommendation was followed (line 319).

- Indeed, Vc and Ve and NAC, are all well-known antioxidants. Have the authors tried to use these antioxidants in the snail experiments?   
It would be interesting to test whether these antioxidants can boost the immune system during experimental dormancy, as shown in the giant African snail Archachatina marginata during heat stress (Iwuozo et al. [7]). However, we have not tried this yet and thought it would be beyond the current aim of this manuscript.

- In my view, I guess that SOD, glutathione level, and metabolite compounds upon ROS or OS, may also be interesting to be measured in the apple snail to draw a better conclusion, in the same experimental conditions.
In the revised version, we have included the determinations of SOD activity and GSH concentration in the blood of animals of both dormancy states (estivation and hibernation).

- In terms of the ROS production as the authors determined, what kinds of ROS were detected in the apple snail? More details?
Several ROS oxidize the probe 2',7'-dichlorodihydrofluorescein (DCFH) to the fluorescent 2',7'-dichlorofluorescein (DCF) (Winterbourn [8]; Murphy et al. [9]). Therefore, we have used it as an indicator of overall ROS production. In the revised version, this is stated in Materials and Methods (lines 187-191).

- Why HIF-1alpha and NRF2 were discussed but excluded in the experiments?  
The reference about REDOX-sensitive transcription factors in Pomacea canaliculata is only in the Introduction in the revised version (lines 77-83).

- Line 241-245. Such as “Geo Mean = 1.0 ± 0.31 ×103”, regarding the numbers and digitals, I am confused.
All geometric mean values are now expressed as 103. This is now stated in Materials and Methods (lines207-212). Regarding digitals, all cytometry values are expressed with two significant digits. We regret for the confusion in this regard.

- “…preparation for oxidative stress (POS)”. OS stands for oxidative stress, maybe O.K. Well, I do not think it is necessary to have “POS” abbreviated for this phrase.
The paragraph has been rewritten. POS stands for “Preparation for Oxidative Stress” and has been widely used since it was first proposed by Hermes-Lima et al. [10]. We have now defined the term (and abbreviation) in the Introduction and used POS consistently throughout the manuscript.

- All “table S1” should be “Table S1”. And the only Table S1 should move from the SI to the main text.
Changed as recommended throughout the manuscript.

- There was no graphical abstract (GA) to give a visible summary for the whole study. GA is hereby requested.
We regret we did not notice it was not optional but mandatory for Metabolites. A graphical abstract is now provided (line 471).

References in the PDF attached archive.

Reviewer 2 Report

reference 11 - add pages and volumen

reference 14. - pages, publisher

reference 15.  - please bold the year of publishing

reference 52. - bold the year, add publisher and pages

reference 56. - bold the year

reference 61. - bold the year

Author Response

Dear Editor,

We thank the reviewers for their comments on the manuscript. We have edited the manuscript to address their concerns as much as possible. The revised version underwent major reorganization, especially the Introduction and Discussion. Therefore, the reviewers’ comments may refer to paragraphs that no longer exist. In our point-by-point responses, we refer to line numbers of the new version.

In addition to the recommended changes, we have made some minor modifications (spelling, grammar, etc.) that are not indicated in the new version.

We have included in the submission a manuscript with the tracking changes tool, so that reviewers can quickly view the modifications.

We hope that our manuscript is now suitable for publication in Metabolites.

Best regards,

Dr. Maximiliano Giraud-Billoud

CONICET-UNCuyo-UNViMe

Reviewer 2
Thank you for noticing the mistakes in the References section (Endnote errors). We have corrected them all (lines 485, 489, 552, 652). Other mistakes referred to references that have been deleted in the revised version of the manuscript.

Reviewer 3 Report

Metabolites

Article: “Short-term estivation and hibernation … snail Pomacea canaliculata

Authors: Rodriguez et al.

In this study, the authors have evaluated the effects of dehydration and cold temperature on the activities of antioxidant enzymes in the circulation and hemocyte response in the apple snail, Pomacea canaliculate. The findings reported in this study augment our knowledge of how the snail adopts strategies to develop tolerance to short-term dehydration and cold temperature. However, I noticed many problems that I have outlined below.

Abstract

I am not sure if natural environment [i.e., in plural form - environments] is the appropriate use.

I am also not sure if the terms estivation and hibernation are appropriate for the apple snail Pomacea canaliculate. Instead, I suggest the word torpor be used, such as summer torpor (estivation) and winter torpor (hibernation).

Does the word quiescence reflect both torpor conditions? To avoid confusion, one should be consistent as regards terms used to define the conditions, such as summer torpor or winter torpor.

Introduction

The first two sentences appear to be complex and confusing. The authors may like to parse those into smaller and more intelligible sentences.

I found the terms, such as dehydration, estivation, hibernation, quiescence, and cold stress to describe short-term stressful conditions, such as dehydration and low temperature.  

Materials and Methods

What are the characteristics of the controlled condition? Instead of giving a citation, the authors should have explained the controlled condition.

What is the limit of temperature tolerance of the snails used in the experiment? Do they form clusters under low or high temperatures?

The glycerol content in cold-exposed snails should have been studied.

Results

The hemocyanin that constitutes more than 90% of all proteins in the molluscan hemolymph should have shown a parallel pattern in total proteins. However, the mean level of total protein in the so-called Arousal-Hib group was the highest. This did not correspond with the mean level of hemocyanin in the same group. This appears probably due to the higher levels of variance among the data sets.

Discussion

A proper explanation for the mismatch between protein levels and hemocyanin levels across the experimental groups has not been discussed adequately.

What is the role of gender vis-à-vis the pattern of variability in circulating hemocytes across various treatment groups? The role of gender in respect of other measured variables has not been discussed adequately. 

Author Response

Dear Editor,

We thank the reviewers for their comments on the manuscript. We have edited the manuscript to address their concerns as much as possible. The revised version underwent major reorganization, especially the Introduction and Discussion. Therefore, the reviewers’ comments may refer to paragraphs that no longer exist. In our point-by-point responses, we refer to line numbers of the new version.

In addition to the recommended changes, we have made some minor modifications (spelling, grammar, etc.) that are not indicated in the new version.

We have included in the submission a manuscript with the tracking changes tool, so that reviewers can quickly view the modifications.

We hope that our manuscript is now suitable for publication in Metabolites.

Best regards,

Dr. Maximiliano Giraud-Billoud

CONICET-UNCuyo-UNViMe

Reviewer 3

Abstract
- I am not sure if natural environment [i.e., in plural form - environments] is the appropriate use.
The Abstract has been rewritten and the phrase no longer exists.

- I am also not sure if the terms estivation and hibernation are appropriate for the apple snail Pomacea canaliculate. Instead, I suggest the word torpor be used, such as summer torpor (estivation) and winter torpor (hibernation).
Does the word quiescence reflect both torpor conditions? To avoid confusion, one should be consistent as regards terms used to define the conditions, such as summer torpor or winter torpor.
The terms ‘estivation’ and ‘hibernation’ have been widely used for ampullariids in general, and, particularly for P. canaliculata, our laboratory has also used both terms since the publication by Giraud-Billoud et al. [11] and thereafter (Giraud-Billoud et al. [1,3,4,12]). ‘Estivation’ refers to dormancy induced by lack of water, and ‘hibernation’ refers to dormancy when the ambient temperature becomes low (as defined in Hayes et al. [13]. We adopt here that terminology and make it clear in the Introduction section (lines 64-66). 

Introduction
- The first two sentences appear to be complex and confusing. The authors may like to parse those into smaller and more intelligible sentences.

The Introduction has been reorganized and rewritten for clarity and conciseness (please see our response to Rev. 1). We have extensively revised the style and tried to make the sentences simpler.

- I found the terms, such as dehydration, estivation, hibernation, quiescence, and cold stress to describe short-term stressful conditions, such as dehydration and low temperature.
We are sorry for the hodgepodge of terms. We now use estivation and hibernation and refer to them as dormant states, as defined by Hayes et al. (2015). We have also checked the consistent use of terms throughout the manuscript.

Materials and Methods
- What are the characteristics of the controlled condition? Instead of giving a citation, the authors should have explained the controlled condition.
The controlled condition is characterized by active animals kept in aquaria at 26–28°C and fed ad libitum a diet composed of fresh lettuce, dried P. canaliculata eggs, and carp food pellets. A full description of the culture conditions is found in Giraud-Billoud et al. [11]. We state this in Materials and Methods (lines 105-109).

- What is the limit of temperature tolerance of the snails used in the experiment? Do they form clusters under low or high temperatures?
The snails used in the experiments have minimum temperature tolerance to 9ºC (as shown in Giraud-Billoud et al. [3]). They form clusters only during the period of experimental hibernation (personal observation). We have not made experiments above 28ºC. Seuffert and Martín [14,15] have reported the thermal limit temperatures for P. canaliculata in its natural habitat. This is now stated in the Introduction (lines 66-67).

- The glycerol content in cold-exposed snails should have been studied.
The Reviewer may be referring here to the role of glycerol as a presumed cryoprotectant, as modestly shown for the heterobranch Helix aspersa (Biannic & Daguzan [16]). Glycerol content has been studied in various terrestrial gastropods and, although its concentration increases in response to low temperatures, its function as a cryoprotectant is less clear (e.g., Nowakowska et al. [17]; Loomis [18]).

Matsukura et al. [19] found that glycerol concentration was indeed increased in overwintering P. canaliculata snails, but the amount of increase was so low that it led the authors to conclude that glycerol content was not sufficient to play a role as a cryoprotectant. This conclusion was supported in further study. Matsukura et al. [20] studied the changes in the glycerol content of cold-acclimated snails and non-acclimated ones. Regardless of whether the snails were acclimated to low temperatures, the supercooling point did not differ from that of non-acclimated ones.

Results
- The hemocyanin that constitutes more than 90% of all proteins in the molluscan hemolymph should have shown a parallel pattern in total proteins. However, the mean level of total protein in the so-called Arousal-Hib group was the highest. This did not correspond with the mean level of hemocyanin in the same group. This appears probably due to the higher levels of variance among the data sets.
Please see our response below.

Discussion
- A proper explanation for the mismatch between protein levels and hemocyanin levels across the experimental groups has not been discussed adequately.
Hemocyanin increased significantly only in the estivation group and returned to the control level in the arousal-est group. Lowry’s technique for measuring protein concentration also detects free amino acids in the blood [21]. Therefore, the increase in hemocyanin may be accompanied by a decrease in amino acids or even other proteins, which can produce a constant total protein concentration.

- What is the role of gender vis-à-vis the pattern of variability in circulating hemocytes across various treatment groups? The role of gender in respect of other measured variables has not been discussed adequately.
In our experiments, we have used a 1:1 sex ratio and found no significant differences between male and female animals.

The references have been included in th PDF attached archive.

Round 2

Reviewer 1 Report

Manuscript ID:  metabolites-2178057                            Type:  Article

Title:  Short-term estivation and hibernation induce changes in the blood and circulating hemocytes of the apple snail Pomacea canaliculate

The manuscript entitled of “Short-term estivation and hibernation induce changes in the blood and circulating hemocytes of the apple snail Pomacea canaliculate” by Cristian Rodriguez et al., was revised and has been re-submitted to Metabolites.

This time, great that the manuscript takes on a new look. Almost all my concerns were properly addressed and all questions were responded. Most importantly, the data presentation and the main texts have been revised in a good manner therefore the quality of the manuscript was improved greatly.

One last suggestion on the revised version of the manuscript is, that Figure S1 could be moved to the main body of the paper. The amounts of data including figures and tables in the main texts are not crowd. The meaning for keeping it as supplementary material is none. Apart from this minor issue, I would suggest an Acceptance on this paper can be made. 

My recommendation for this manuscript in its current form is: to be Accepted.

Reviewer 3 Report

No comments on the revised manuscript.